

# Precision-machine learning for the matrix element method

Theo Heimel[1], Nathan Huetsch[1], Ramon Winterhalder[2],
Tilman Plehn[1] and Anja Butter[1,3]

**1** Institut für Theoretische Physik, Universität Heidelberg, Germany
**2** CP3, Université catholique de Louvain, Louvain-la-Neuve, Belgium
**3** LPNHE, Sorbonne Université, Université Paris Cité,
CNRS/IN2P3, Paris, France

## Abstract

The matrix element method is the LHC inference method of choice for limited statistics. We present a dedicated machine learning framework, based on efficient phase-space integration, a learned acceptance and transfer function. It is based on a choice of INN and diffusion networks, and a transformer to solve jet combinatorics. We showcase this setup for the CP-phase of the top Yukawa coupling in associated Higgs and single-top production.

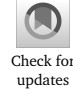
# 1 Introduction and reference process

Optimal analyses are the key challenge for the current and future LHC program, including specific model-based as well simulation-based search strategies. A classic method is the matrix element method (MEM), developed for the top physics program at the Tevatron [1, 2]. It derives its optimality from the Neyman-Pearson lemma and the fact that all information for a given hypothesis is encoded in the differential cross section. In the MEM, we compute likelihood ratios for individual events, such that the log-likelihood ratio of an event sample is the sum of the event-wise log-likelihood ratios. A combination of events to a kinematic distribution is not necessary [3].

The MEM was first used in the top mass measurement [4–7] and the discovery of the single-top production process [8] at the Tevatron. At the LHC, there exist several studies [9–15] and analysis applications [14, 16–19]. The critical challenge to MEM analyses is the integration over all possible parton-level configurations which could lead to the analyzed observed events. It can be solved by using modern machine learning (ML) for a fast and efficient combination of simulation and integration [20, 21]. A related ML approach to likelihood extraction is the classifier-based estimation of likelihood ratios [22].

We present a comprehensive simulation and integration framework for the MEM, based on modern machine learning [23, 24]. It makes extensive use of generative networks, which are transforming LHC simulations and analyses just like any other part of our lives. This starts with phase-space integration and sampling [25–30] and continues with more LHC-specific tasks like event subtraction [31], event unweighting [32, 33], loop integrations [34], or super-resolution enhancement [35, 36]. At the LHC, generative networks generally work in interpretable physics phase spaces, for example scattering events [37–43], parton showers [44–51], and detector simulations [52–75]. These networks can be trained on first-principle simulations and are easy to handle, efficient to ship, powerful in amplifying the training samples [76, 77], and — most importantly — precise [42, 78–80]. Conditional versions of these established generative networks then enable new analysis methods, like probabilistic unfolding [81–89], inference [21, 90], or anomaly detection [91–96].

We introduce a new MEM-ML-analysis framework in Sec. 2. It combines two generative network and one classifier network and pushes the precision beyond our conceptual study [21], towards an experimentally required level. For a fast and bi-directional evaluation we use the established cINNs with advanced coupling layers [42], updated to current precision requirements in Sec. 3. In Sec. 4, we add a learned acceptance network. In Sec. 5, we show how a generative diffusion network [43] improves the precision, albeit at the expense of speed. Finally, we employ a transformer architecture [43, 97, 98] to solve the jet combinatorics in Sec. 6. This series of improvements allows us to extract likelihood distributions from small and moderate-size event samples without a network bias and with close-to-optimal performance.

**Reference process**

The focus of this paper is entirely on our new ML-method to enable MEM analyses at the LHC. However, we use an established, challenging, and realistic physics process to illustrate our method. This reference process is introduced and discussed in Ref. [21]. We target the purely hadronic signature

$$pp \to tHj \to (bjj)(\gamma\gamma)j + \text{jets}, \tag{1}$$

with up to four additional jets from QCD radiation. The production process allows for a measurement of a CP-phase in the top Yukawa coupling at future LHC runs [99–107]. The challenge of having to work with small event numbers is motivated by choosing the rare decay

channel $H \to \gamma\gamma$, which allows us to control continuum backgrounds efficiently. The total cross section is 43.6 fb, when we combine top and anti-top production.

To probe the symmetry structure of the Yukawa coupling, we introduce a mixed CP-even and CP-odd interaction [108],

$$\mathcal{L}_{t\bar{t}H} = -\frac{y_t}{\sqrt{2}}\Big[a\cos\alpha\,\bar{t}t + ib\sin\alpha\,\bar{t}\gamma_5 t\Big]H\,. \tag{2}$$

Choosing $a = 1$ and $b = 2/3$ [109] keeps the cross section for $gg \to H$ constant. The model parameter we target with the matrix element method is the CP-angle $\alpha$. For more details on this reference process we refer to our conceptual study [21]. Obviously, all our findings can be generalized to other LHC processes.

## 2 ML-matrix element method

The matrix element method is a simulation-based inference method which uses the fact that for a given parameter of interest, $\alpha$, the likelihood can be extracted from a simulation of the differential cross section. It describes the hard scattering process and factorizes into the total cross section and a normalized probability density,

$$\frac{d\sigma(\alpha)}{dx_{\text{hard}}} = \sigma(\alpha)\,p(x_{\text{hard}}|\alpha) \qquad \Leftrightarrow \qquad p(x_{\text{hard}}|\alpha) = \frac{1}{\sigma(\alpha)}\frac{d\sigma(\alpha)}{dx_{\text{hard}}}\,. \tag{3}$$

Given the hard process, we then simulate the parton shower, hadronization, detector effects, and the reconstruction of analysis objects, with a forward-transfer or response function $r$ [110]. This function is assumed to be independent of the theory parameter $\alpha$

$$x_{\text{hard}} \quad\xrightarrow{\quad r(x_{\text{reco}}|x_{\text{hard}})\quad} x_{\text{reco}} \atop \xrightarrow[\quad p_{\text{reject}}(x_{\text{hard}})\quad]{} \text{rejected.} \tag{4}$$

The detector geometry and acceptance cuts will lead to, either, a valid reco-level event $x_{\text{reco}}$ or a rejected event, introducing $p_{\text{reject}}(x_{\text{hard}})$ as the probability that a given hard event $x_{\text{hard}}$ is rejected. The transfer function $r$ is not normalized, and a proper normalization condition defines the efficiency or acceptance function,

$$\epsilon(x_{\text{hard}}) := \int dx_{\text{reco}}\,r(x_{\text{reco}}|x_{\text{hard}}) = 1 - p_{\text{reject}}(x_{\text{hard}})\,. \tag{5}$$

Using the transfer function we can parametrize the forward evolution of the differential cross section following

$$\frac{d\sigma_{\text{fid}}(\alpha)}{dx_{\text{reco}}} = \int dx_{\text{hard}}\,r(x_{\text{reco}}|x_{\text{hard}})\frac{d\sigma(\alpha)}{dx_{\text{hard}}}\,, \tag{6}$$

where the subscript 'fid' indicates that the reco-level phase space is different from the parton level. In this relation we can use Eq.(5) to replace $r$ with a normalized transfer probability $p(x_{\text{reco}}|x_{\text{hard}})$,

$$r(x_{\text{reco}}|x_{\text{hard}}) = \epsilon(x_{\text{hard}})\,p(x_{\text{reco}}|x_{\text{hard}})\,, \qquad \text{with} \qquad \int dx_{\text{reco}}\,p(x_{\text{reco}}|x_{\text{hard}}) = 1\,. \tag{7}$$

Inserting Eq.(7) in Eq.(6) we obtain the final expression for the differential cross section

$$\frac{d\sigma_{\text{fid}}(\alpha)}{dx_{\text{reco}}} = \int dx_{\text{hard}}\,\epsilon(x_{\text{hard}})\,p(x_{\text{reco}}|x_{\text{hard}})\frac{d\sigma(\alpha)}{dx_{\text{hard}}}\,. \tag{8}$$

Equivalent to Eq.(3) we can now define the likelihood for reco-level events in terms of the fiducial cross section and the differential cross section

$$\frac{d\sigma_{\text{fid}}(\alpha)}{dx_{\text{reco}}} = \sigma_{\text{fid}}(\alpha)\, p(x_{\text{reco}}|\alpha) \qquad \Leftrightarrow \qquad p(x_{\text{reco}}|\alpha) = \frac{1}{\sigma_{\text{fid}}(\alpha)}\frac{d\sigma_{\text{fid}}(\alpha)}{dx_{\text{reco}}}. \tag{9}$$

To obtain the fiducial cross section $\sigma_{\text{fid}}(\alpha)$, we now need to integrate Eq.(8) over the reco-level phase space

$$\begin{aligned}
\sigma_{\text{fid}}(\alpha) &= \int dx_{\text{reco}} \int dx_{\text{hard}}\, \epsilon(x_{\text{hard}})\, p(x_{\text{reco}}|x_{\text{hard}})\frac{d\sigma(\alpha)}{dx_{\text{hard}}} \\
&= \int dx_{\text{hard}}\, \epsilon(x_{\text{hard}})\frac{d\sigma(\alpha)}{dx_{\text{hard}}} \\
&= \sigma(\alpha)\int dx_{\text{hard}}\, \epsilon(x_{\text{hard}})\, p(x_{\text{hard}}|\alpha) \\
&= \sigma(\alpha)\big\langle \epsilon(x_{\text{hard}})\big\rangle_{x\sim p(x_{\text{hard}}|\alpha)},
\end{aligned} \tag{10}$$

where we first use Eq.(7) to integrate out the reco-level phase space and then replace the differential cross section using Eq.(3). This allows us to express the integral in terms of the average acceptance $\langle\epsilon\rangle_\alpha$ which is used to evaluate the integral numerically. Using Eq. (8) in Eq. (9) we obtain the final expression for the reco-level likelihood

$$p(x_{\text{reco}}|\alpha) = \frac{1}{\sigma_{\text{fid}}(\alpha)}\int dx_{\text{hard}}\frac{d\sigma(\alpha)}{dx_{\text{hard}}}\,\epsilon(x_{\text{hard}})\, p(x_{\text{reco}}|x_{\text{hard}}). \tag{11}$$

Note that in our training dataset, consisting of simulated event pairs $(x_{\text{reco}}, x_{\text{hard}})$, the hard-scattering momenta are not distributed according to Eq.(3), because it does not contain events $x_{\text{hard}}$ that have been rejected. Consequently, the accepted $x_{\text{hard}}$ are distributed as

$$p_{\text{fid}}(x_{\text{hard}}|\alpha) = \frac{1}{\sigma_{\text{fid}}(\alpha)}\frac{d\sigma(\alpha)}{dx_{\text{hard}}}\,\epsilon(x_{\text{hard}}). \tag{12}$$

This means, we can directly relate the reco-level likelihood to a modified parton-level likelihood

$$p(x_{\text{reco}}|\alpha) = \int dx_{\text{hard}}\, p(x_{\text{reco}}|x_{\text{hard}})\, p_{\text{fid}}(x_{\text{hard}}|\alpha), \tag{13}$$

which connects the MEM with the completeness relation from statistics.

**Acceptance classifier and transfer network**

To compute the reco-level likelihood defined in Eq.(11) we rely on $\epsilon(x_{\text{hard}})$ and $p(x_{\text{reco}}|x_{\text{hard}})$, defined through a forward simulation. We encode both functions in neural networks trained on these forward simulations.

First, the acceptance $\epsilon(x_{\text{hard}})$ can be encoded as a standard classifier network

$$x_{\text{hard}} \xrightarrow{\text{Acceptance network}} \epsilon_\psi(x_{\text{hard}}), \tag{14}$$

where $\psi$ denotes the trainable network parameters. Given the input $x_{\text{hard}}$ it learns the labels 1 for accepted events and 0 otherwise. Because the network is a classifier with a cross entropy loss, its output will be the acceptance probability for the given event.

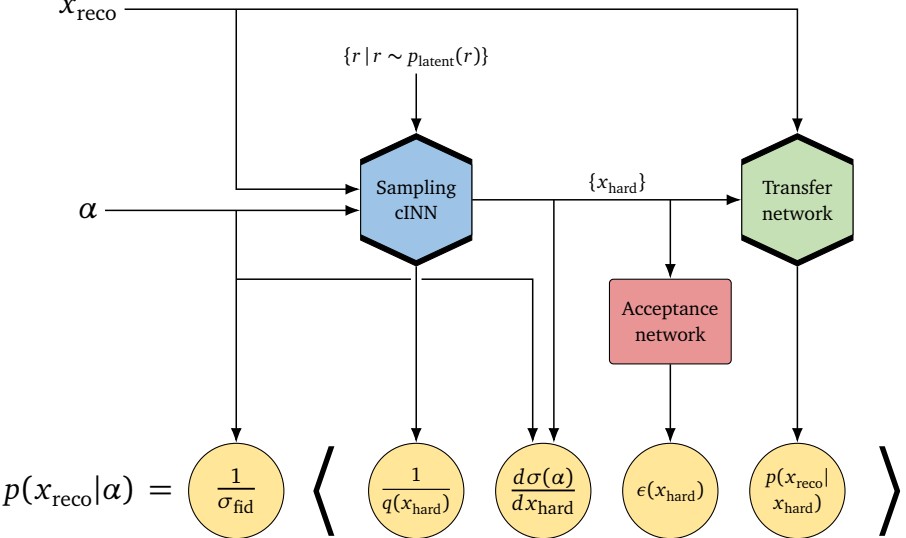

Figure 1: Three-network MEM integrator evaluating Eq.(23) through sampling $r$. The Sampling-cINN is conditioned on the CP-angle $\alpha$ and the reco-level event $x_{\text{reco}}$. The Transfer network is conditioned on the hard-scattering event $x_{\text{hard}}$. For the three-network setup the acceptance $\epsilon(x_{\text{hard}})$ is encoded in a network.

The transfer probability introduced in Eq.(7) is encoded in a generative network with density estimation capability, like a normalizing flow or diffusion model, and is trained on event pairs $(x_{\text{reco}}, x_{\text{hard}})$. For this training dataset, we only include accepted events. The generative network defines a bijective mapping between Gaussian random numbers and reco-level phase space conditioned on parton-level events,

$$x_{\text{reco}} \sim p_\theta(x_{\text{reco}}|x_{\text{hard}}) \quad \xleftrightarrow{\text{Transfer network}} \quad r \sim p_{\text{latent}}(r), \tag{15}$$

with trainable parameters $\theta$. This mapping can than be used for density estimation in the forward direction and for conditional generation of reco-level events in the inverse direction.

**Sampling-cINN**

The integration in Eq.(13) is challenging, because the differential cross section spans several orders of magnitude, and the transfer probability typically forms a narrow peak. We solve the integral using Monte Carlo integration sampling $x_{\text{hard}} \sim q(x_{\text{hard}}|x_{\text{reco}}, \alpha) \equiv q(x_{\text{hard}})$,

$$\begin{aligned} p(x_{\text{reco}}|\alpha) &= \int dx_{\text{hard}} \, p_{\text{fid}}(x_{\text{hard}}|\alpha) \, p_\theta(x_{\text{reco}}|x_{\text{hard}}) \\ &= \left\langle \frac{1}{q(x_{\text{hard}})} p_{\text{fid}}(x_{\text{hard}}|\alpha) \, p_\theta(x_{\text{reco}}|x_{\text{hard}}) \right\rangle_{x_{\text{hard}} \sim q(x_{\text{hard}})}, \end{aligned} \tag{16}$$

Ideally, this assumes

$$p_\theta(x_{\text{reco}}|x_{\text{hard}}) = p(x_{\text{reco}}|x_{\text{hard}}), \tag{17}$$

in which case we can use Bayes' theorem to arrive at

$$p(x_{\text{reco}}|\alpha) = \left\langle \frac{1}{q(x_{\text{hard}})} p_{\text{fid}}(x_{\text{hard}}|\alpha) p(x_{\text{reco}}|x_{\text{hard}}) \right\rangle_{x_{\text{hard}} \sim q(x_{\text{hard}})}$$

$$= \left\langle \frac{1}{q(x_{\text{hard}})} p(x_{\text{hard}}|x_{\text{reco}}, \alpha) p(x_{\text{reco}}|\alpha) \right\rangle_{x_{\text{hard}} \sim q(x_{\text{hard}})}. \tag{18}$$

For this integral the variance vanishes when

$$q(x_{\text{hard}}) \equiv q(x_{\text{hard}}|x_{\text{reco}}, \alpha) \propto p(x_{\text{hard}}|x_{\text{reco}}, \alpha), \tag{19}$$

where $p(x_{\text{hard}}|x_{\text{reco}}, \alpha)$ corresponds to the generative unfolding probability from reco-level to parton-level [84]. However, in practice, we cannot expect the learned transfer probability to match its truth counterpart perfectly. In that case the condition in Eq.(19) becomes

$$q(x_{\text{hard}}|x_{\text{reco}}, \alpha) \propto p_{\text{fid}}(x_{\text{hard}}|\alpha) p_\theta(x_{\text{reco}}|x_{\text{hard}}). \tag{20}$$

In both cases, we train a second conditional normalizing flow with trainable parameters $\varphi$ to encode this optimal transformation of the integration variables,

$$r \sim p_{\text{latent}}(r) \quad \xleftrightarrow{\text{Sampling-cINN}} \quad x_{\text{hard}}(r) \sim q_\varphi(x_{\text{hard}}|x_{\text{reco}}, \alpha), \tag{21}$$

which allows to parameterize the conditional sampling density as

$$q_\varphi(x_{\text{hard}}|x_{\text{reco}}, \alpha) \equiv q_\varphi(x_{\text{hard}}(r)|x_{\text{reco}}, \alpha) = \frac{p_{\text{latent}}(r)}{J_\varphi(r)},$$

$$\text{with} \quad J_\varphi(r) = \left| \frac{\partial x_{\text{hard}}(r; x_{\text{reco}}, \alpha; \varphi)}{\partial r} \right|. \tag{22}$$

The MEM integral in Eq.(11) now reads

$$p(x_{\text{reco}}|\alpha) = \frac{1}{\sigma_{\text{fid}}(\alpha)} \int dr \, J_\varphi(r) \left[ \frac{d\sigma(\alpha)}{dx_{\text{hard}}} \epsilon_\psi(x_{\text{hard}}) p_\theta(x_{\text{reco}}|x_{\text{hard}}) \right]_{x_{\text{hard}}(r; x_{\text{reco}}, \alpha; \varphi)}$$

$$= \frac{1}{\sigma_{\text{fid}}(\alpha)} \left\langle \frac{J_\varphi(r)}{p_{\text{latent}}(r)} \left[ \frac{d\sigma(\alpha)}{dx_{\text{hard}}} \epsilon_\psi(x_{\text{hard}}) p_\theta(x_{\text{reco}}|x_{\text{hard}}) \right]_{x_{\text{hard}}(r; x_{\text{reco}}, \alpha; \varphi)} \right\rangle_{r \sim p(r)}. \tag{23}$$

The architecture of our MEM integrator is illustrated in Fig. 1.

## 3  Two-network baseline

In the proof-of-concept implementation of Ref. [21] we used a series of ad-hoc fixes to stabilize the critical phase space integration in Eq.(11). Before we present more substantial improvements to our framework, we introduce a series of numerical improvements to our baseline two-cINN setup. For the two-network setup we assume that we can neglect the phase-space dependence of the acceptance in the MEM integration,

$$p(x_{\text{reco}}|\alpha) \approx \frac{1}{\sigma_{\text{fid}}(\alpha)} \int dx_{\text{hard}} \frac{d\sigma(\alpha)}{dx_{\text{hard}}} p_\theta(x_{\text{reco}}|x_{\text{hard}}). \tag{24}$$

**Single-pass integration over model parameters**

Initially, we integrate over the phase space for each theory parameter value separately. This general approach does not make use of the fact that the detector response does not depend on $\alpha$, and the mapping for the importance sampling only has a small $\alpha$-dependence. The phase space samples $x_\text{hard} \sim q_\varphi(x_\text{hard}|x_\text{reco}, \alpha)$ and the corresponding values of $p_\theta(x_\text{reco}|x_\text{hard})$ can be used to evaluate the differential cross section for multiple points in $\alpha$. Moreover, parts of the cross section calculation only depend on the phase space point and not on $\alpha$, like for example parton densities.

Consequently, we can understand the integrand for a given Monte Carlo sample as a smooth function of $\alpha$, so the integral will also be a smooth function of $\alpha$. This means we do not have to fit an explicit function to the likelihood values and instead extract a smooth log-likelihood as a function of $\alpha$. The MEM integration for a given $x_\text{reco}$ and a discrete set $\{\alpha\}$ can be performed as:

1. For $j \in \{1, \ldots, N\}$, draw $\alpha^{(j)}$ from $\{\alpha\}$ randomly.

2. Using the sampling network, sample $x_\text{hard}^{(j)} \sim q_\varphi(x_\text{hard}|x_\text{reco}, \alpha^{(j)})$.

3. Evaluate the transfer probability $p_\theta(x_\text{reco}|x_\text{hard}^{(j)})$ for each sample.

4. Evaluate the differential cross section $d\sigma(\alpha)/dx_\text{hard}^{(j)}$ for each sample $x_\text{hard}^{(j)}$ and $\alpha$.

5. Compute the MC integral Eq.(24) for all $\alpha$ values at the same time

$$p(x_\text{reco}|\alpha) \approx \frac{1}{\sigma_\text{fid}(\alpha)} \frac{1}{N} \sum_{j=1}^{N} \frac{1}{q_\varphi(x_\text{hard}^{(j)}|x_\text{reco}, \alpha^{(j)})} \frac{d\sigma(x_\text{hard}^{(j)}|\alpha)}{dx_\text{hard}} p_\theta(x_\text{reco}|x_\text{hard}^{(j)}). \quad (25)$$

This integral converges quickly for some events, while more statistics are needed for others. One reason is that the peaks of the transfer probability and the importance sampling distribution are not perfectly aligned for some events, resulting in a higher variance. To reduce the integration time while guaranteeing a small integration error, we compute the integral iteratively. We specify the number of samples per iteration as well as a minimal and maximal number of iterations. Furthermore, we specify a threshold for the maximum relative uncertainty over the results for all values of $\alpha$. The integration is repeated for new batches of samples until the combined uncertainty drops below the threshold. In practice, a batch size of 10000, at least two and at most 15 iterations meet a target uncertainty of 2%. The uncertainty on the normalized negative log-likelihood will be much smaller than these 2% because of the correlation between different $\alpha$.

**Integration uncertainties**

Using this single-pass integration, the results for different $\alpha$ values become correlated, because the new algorithm ensures that the result is a smooth function of $\alpha$. This means that the MC integration error cannot be easily estimated point-wise. The uncertainty on the likelihood ratio should be much smaller than the uncertainty of the absolute value of the likelihood before normalization. To account for the correlations, we use bootstrapping to resample the integrand multiple times and propagate the resulting replicas through the downstream tasks. For this bootstrapping we take our samples of the integrand $I^{(j)}(\alpha_i)$ and randomly draw $M$ batches of $N$ samples from $\{I^{(j)}(\alpha_i) \,|\, j \in \{1, \ldots, N\}\}$ with replacement. We compute the mean over the $N$ samples per batch, defining $M$ replicas of the integral as a function of $\alpha$. They can be used to estimate uncertainties on the following normalized negative log-likelihoods.

Next, we can quantify the uncertainty from the training of the transfer probability using a Bayesian network [111–117]. To estimate the training uncertainty we perform the phase space integration for different samples from the distribution over the trainable parameters. In Ref. [21] this is done by repeating the integration for different sampled networks. However, the idea of the single-pass integration also applies to the Bayesian transfer probabilities. The same importance sampling distribution should work well for different sampled networks, making the integration more efficient. The training uncertainty estimation can be combined with the bootstrapping procedure described above. For each replica, we do not only resample the integrand but also compute the transfer probability for a different sample from the distribution over the trainable parameters.

**Factorization of differential cross section**

For our example process, single-top plus Higgs production with an anomalous CP-phase, the Lagrangian given in Eq.(2) can be written as

$$\mathcal{L} = \mathcal{L}_1 + \sin\alpha\,\mathcal{L}_2 + \cos\alpha\,\mathcal{L}_3\,, \tag{26}$$

and the squared matrix element has the corresponding form

$$\frac{d\sigma(x_\text{hard}|\alpha)}{dx_\text{hard}} = g_1 + \sin\alpha\,g_2 + \cos\alpha\,g_3 + \sin\alpha\cos\alpha\,g_4 + \sin^2\alpha\,g_5\,, \tag{27}$$

with phase space dependent $g_i(x_\text{hard})$. This is an example where the matrix element factorizes into an $x_\text{hard}$-dependent and an $\alpha$-dependent part. Similar factorization properties hold for SMEFT corrections where it is often referred to as operator morphing [118]. For

$$\frac{d\sigma(x_\text{hard}|\alpha)}{dx_\text{hard}} = \sum_i f_i(\alpha)g_i(x_\text{hard})\,, \tag{28}$$

the MEM integration in Eq.(24) becomes

$$p(x_\text{reco}|\alpha) = \frac{1}{\sigma_\text{fid}(\alpha)}\sum_i f_i(\alpha)\int dx_\text{hard}\,g_i(x_\text{hard})p_\theta(x_\text{reco}|x_\text{hard})\,. \tag{29}$$

The same can be done for the Monte Carlo estimate of the integral,

$$p(x_\text{reco}|\alpha) \approx \frac{1}{\sigma_\text{fid}(\alpha)}\frac{1}{N}\sum_{j=1}^{N}\frac{1}{q_\varphi(x_\text{hard}^{(j)}|x_\text{reco},\alpha^{(j)})}\frac{d\sigma(x_\text{hard}^{(j)}|\alpha)}{dx_\text{hard}}p_\theta(x_\text{reco}|x_\text{hard}^{(j)}) \tag{30}$$

$$= \frac{1}{\sigma_\text{fid}(\alpha)}\sum_i f_i(\alpha)\frac{1}{N}\sum_{j=1}^{N}\frac{1}{q_\varphi(x_\text{hard}^{(j)}|x_\text{reco},\alpha^{(j)})}g_i(x_\text{hard}^{(j)})p_\theta(x_\text{reco}|x_\text{hard}^{(j)})\,, \tag{31}$$

where $x_\text{hard}^{(j)}\sim q_\varphi(x_\text{hard}|x_\text{reco},\alpha^{(j)})$. The exact functional form of the integral is only preserved if the same $x_\text{hard}^{(j)}$ are used for all values of $\alpha$.

**Importance sampling trained on transfer probability**

The training of the Sampling-cINN assumes that the transfer network encodes $p(x_\text{reco}|x_\text{hard})$ perfectly. The Sampling-cINN is then used for importance sampling. From that perspective, it is less important to learn the truth distribution

$$q_\varphi(x_\text{hard}|x_\text{reco},\alpha) \approx p(x_\text{hard}|x_\text{reco},\alpha) \propto p(x_\text{reco}|x_\text{hard})p_\text{fid}(x_\text{hard}|\alpha)\,, \tag{32}$$



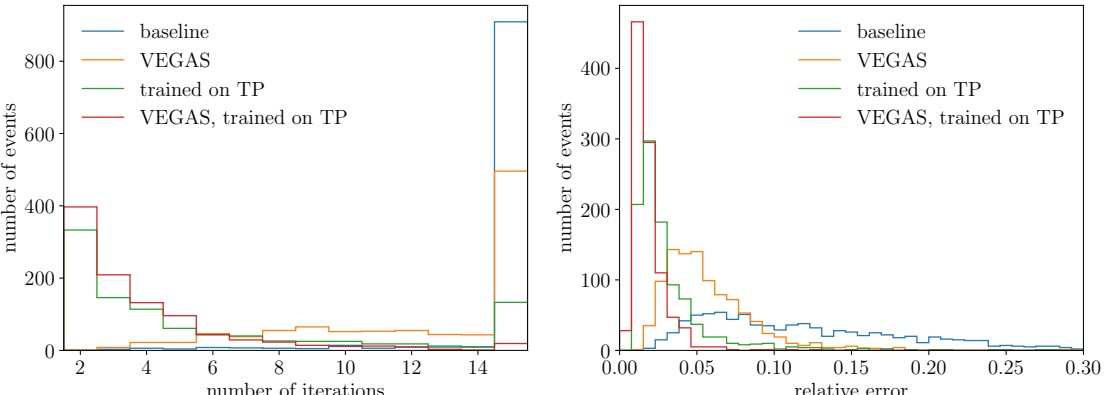

Figure 2: Integration performance with and without importance sampling trained on the transfer probability and VEGAS refinement. Left: number of iterations (10000 samples each) to reach the 2% target precision, with 2 to 15 iterations. Right: relative integration error after 10 iterations of 10000 samples each.

than the modeled distribution

$$q_\varphi(x_{\text{hard}}|x_{\text{reco}}, \alpha) \approx p_\theta(x_{\text{reco}}|x_{\text{hard}})p_{\text{fid}}(x_{\text{hard}}|\alpha). \qquad (33)$$

The training data, consisting of tuples $(\alpha, x_{\text{hard}}, x_{\text{reco}})$ should then be modified by replacing the reco-level momentum with the generated $\tilde{x}_{\text{reco}} \sim p_\theta(x_{\text{reco}}|x_{\text{hard}})$. To increase the training statistics we re-sample the reco-level momenta at the beginning of each epoch. Because of the sharply peaked form of the transfer probability, even small deviations from the truth that do not have a significant impact on the inference performance, can lead to a significant misalignment with the importance sampling distribution. Hence, training the importance sampling on the learned transfer probability leads to a significantly better variance of the integrations weights and a faster convergence of the integral.

### VEGAS latent space refinement

Even when the Sampling-cINN is trained on the learned transfer probability, some events lead to a large variance in the MEM integration. This can be solved by further adapting the proposal distribution during the integration. Specializing the importance sampling network for such an event is impracticable. An alternative is to refine the INN latent space using VEGAS. Instead of directly sampling random numbers and mapping them to phase space, we transform them with a VEGAS grid first. Note, that the grid is shared for all $\alpha$ because of the small $\alpha$ dependence of the importance sampling. After each iteration of the integration, this grid is adapted to reduce the variance of the integral. Because we need to pass the integrand value back to VEGAS, we choose a value in the middle of the relevant $\alpha$-interval being evaluated. The results from the different iterations of the integrals are combined by weighting them by the inverse variance to reduce the overall variance and especially the effect of early iterations where the grid is not yet well adapted.

Figure 2 illustrates the effect of training the Sampling-cINN on the transfer probability and using VEGAS refinement for the MEM integration performance with 1000 SM events and networks with a similar architecture and hyperparameters as in Ref. [21]. For our baseline, we use single-pass integration including a factorized differential cross section. While this guarantees smooth likelihood curves as a function of $\alpha$, we find that the integration uncertainty does not meet the target precision of 2% within 15 iteration for most events. Running the integration with VEGAS refinement improves the convergence, and the importance sampling

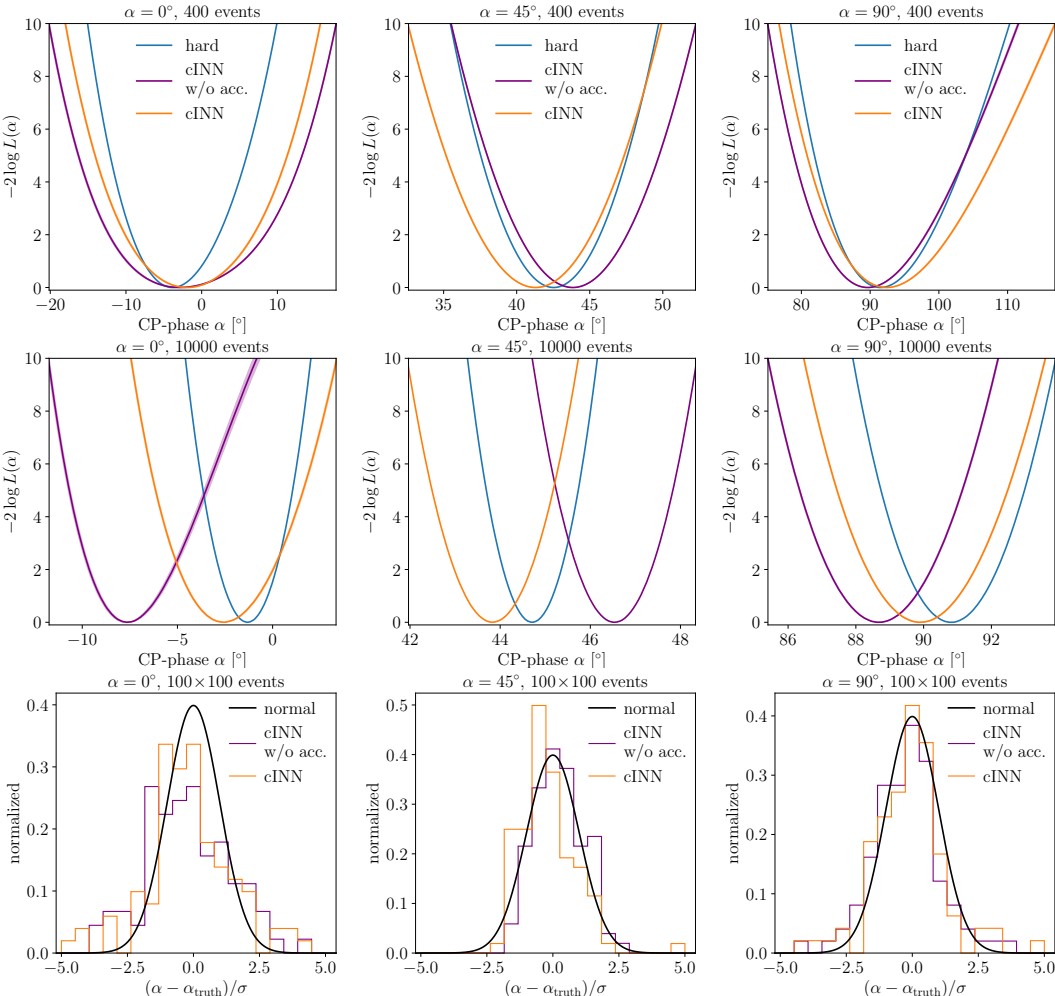

Figure 3: **cINN benchmark and learned acceptance:** likelihoods for different CP-angles. We use the same architecture as in Ref. [21], but with the improved integration. The purple curve shows the two-network cINN benchmark and the orange curve also includes the learned acceptance. From top to bottom: likelihoods for 400 events, 10000 events, and pulls.

trained on the transfer probability leads to a even larger improvements. The combination of both methods ensures that the target precision is reached within 15 iterations for most events. This shows that the Sampling-cINN, trained on the transfer function and with VEGAS refinement, appears to be sufficiently precise to ensure fast convergence of the phase space integral.

**Two-network cINN benchmark**

The purple line in Fig. 3 shows the extracted log-likelihoods for our example process, using all improvements described in this section, and similar architecture and hyperparameters as in Ref. [21]. In the top two rows we show the extracted likelihoods from a small set of 400 events and from a large set of 10k events. In both cases, we compare the likelihood extracted from the reconstructed events to the hard-process truth. Note that we show the integration uncertainties as error bands in the plots, but due to our low error threshold and the single-pass integration these are barely visible. By repeating the integration with the same networks, we confirm that the result is perfectly stable and consistent with these uncertainties.

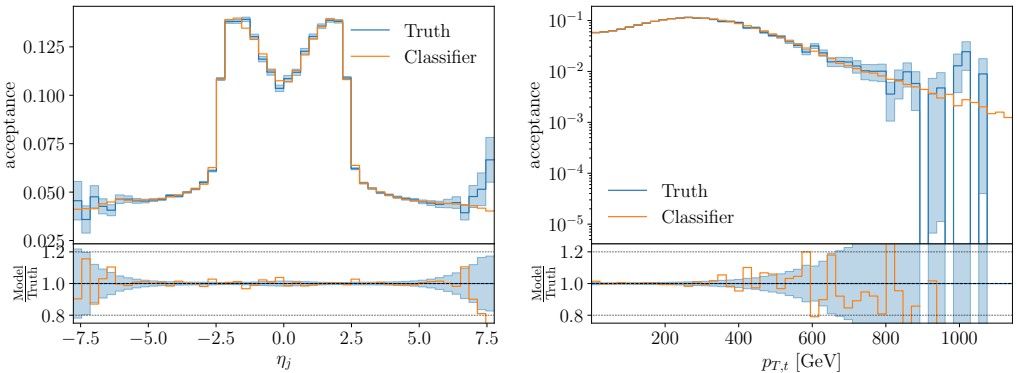

Figure 4: Truth (dashed line) and learned (solid line) acceptance as a function of different kinematic observables.

Performance issues occur when we increase the number of events. The precision of the combined likelihood increases and leads to a systematic deviation between the hard-process and the reconstructed likelihoods. This is not caused by the integration, and we will target this shortcoming by improving the architecture and the training of the transfer probability.

# 4 Acceptance classifier

Moving from the two-network setup in Sec. 3 to the new, three-network setup introduced in Sec. 2, we are back to the more general form of the MEM-integral,

$$p(x_{\text{reco}}|\alpha) = \frac{1}{\sigma_{\text{fid}}(\alpha)} \int dx_{\text{hard}} \frac{d\sigma(\alpha)}{dx_{\text{hard}}} \epsilon_\psi(x_{\text{hard}}) p_\theta(x_{\text{reco}}|x_{\text{hard}}). \tag{34}$$

The acceptance function will be encoded in a straightforward classifier network. It targets the scenario where the jet from the hard process escapes detection, i.e. $|\eta_j| > 2.4$, while the event is still accepted since a ISR jet is tagged instead. The two possible origins of the jets are taken into account by the transfer probability. An additional challenge is the significant drop in acceptance which is now remedied automatically by the introduction of the classifier to include the acceptance rate.

We train the classifier on a dataset of hard process configurations with the additional information of the acceptance label. Its output then provides $\epsilon_\psi(x_{\text{hard}})$ to solve Eq.(34). Its hyperparameters are given in Tab. 1, and its training only takes a few minutes. The learned and true acceptances as a function of different kinematic observables are shown in Fig. 4. Indeed, we see a large jump in the acceptance at $|\eta_j| = 2.4$, by almost a factor three. Also for other observables, like the $p_T$ of the top, the acceptance varies considerably over phase space.

We then evaluate the MEM integral, now including the learned acceptance. Comparing the new results (orange) with the two-network baseline (purple) in Fig. 3 we see a considerable improvement. For the small set of 400 events there is no bias left between the extracted likelihoods and the hard-process truth. Also for 10k events the large bias from Fig. 3 is reduced to a level where it is comparable to the statistical precision. Even for the challenging SM-case $\alpha = 0°$ the extracted likelihoods agrees well with the truth extracted from the hard process. The remaining question is how close we can bring the widths of the extracted likelihood-curves to the optimal outcome from the hard process, and if a remaining systematic bias can keep up with statistical improvements. From now on, we will keep the acceptance network within our MEM setup throughout the rest of our paper.

# 5 Transfer diffusion

Instead of a Transfer-cINN [21], as discussed in Sec. 2, we can also use other neural networks to encode the transfer probability. The great advantages of the INN are its stability, its controlled precision in estimating the density, and its speed in both directions. However, these advantages come at the prize of limited flexibility, and we can use diffusion networks to slightly shift this balance [43]. Conditional flow matching (CFM) networks [119–121] allow for more flexibility in encoding an underlying density, with the main disadvantage of a significant loss in speed in the likelihood evaluation. While this speed might become a relevant factor eventually, we compare the performance of the cINN with the CFM at face value. For a detailed introduction of conditional flow matching in the context of particle physics we refer to Ref. [43] and only repeat the key points here.

The Transfer-CFM replaces the Transfer-cINN in Eq.(15). The CFM models the transformation between a latent distribution $p_{\text{latent}}(r)$ and a conditional phase space distribution $p_\theta(x_{\text{reco}}|x_{\text{hard}})$ inspired by a a time-dependent process. The time evolution is described by an ordinary differential equation

$$\frac{dx(t)}{dt} = v(x(t), t),\qquad(35)$$

with the velocity field $v(x(t), t)$. The corresponding time-dependent probability density $p(x, t)$ obeys the continuity equation

$$\frac{\partial p(x, t)}{\partial t} + \nabla_x [p(x, t)v(x, t)] = 0.\qquad(36)$$

To obtain a generative model we need a velocity field that evolves the probability density in time such that

$$p(x, t) \rightarrow \begin{cases} p_\theta(x) \approx p_{\text{data}}(x), & t \rightarrow 0, \\ p_{\text{latent}}(x) = \mathcal{N}(x; 0, 1), & t \rightarrow 1. \end{cases}\qquad(37)$$

To construct this velocity field we start from a sample-conditional diffusion trajectory

$$x(t|x_0) = (1-t)x_0 + tr \rightarrow \begin{cases} x_0, & t \rightarrow 0, \\ r \sim \mathcal{N}(0, 1), & t \rightarrow 1, \end{cases}\qquad(38)$$

that evolves the phase space sample $x_0$ towards a latent space sample. The associated sample-conditional velocity field directly follows from the ODE Eq.(35)

$$v(x(t|x_0), t|x_0) = \frac{d}{dt}[(1-t)x_0 + tr] = -x_0 + r.\qquad(39)$$

The desired velocity field for the generative model is then given by [119]

$$v(x, t) = \int dx_0 \frac{v(x, t|x_0)p(x, t|x_0)p_{\text{data}}(x_0)}{p(x, t)}.\qquad(40)$$

Learning the velocity field from data is a straightforward regression task and can again be reformulated in terms of the conditional velocity field [119]

$$\mathcal{L}_{\text{FM}} = \left\langle [v_\theta(x, t) - v(x, t)]^2 \right\rangle_{t, x \sim p(x, t)}$$

$\Big\downarrow$ reparametrization + neglecting constants

$$\mathcal{L}_{\text{CFM}} = \left\langle [v_\theta(x(t|x_0), t) - v(x(t|x_0), t|x_0)]^2 \right\rangle_{t \sim U(0,1), x_0 \sim p_{\text{data}}, r \sim \mathcal{N}(0,1)}.\qquad(41)$$

Once the model is trained to encode the velocity it defines a bijective mapping between the latent and the phase space via numerically solving the ODE Eq.(35). Crucially for our application the Jacobian of this transformation is tractable through another ODE [122]

$$\frac{d\log p(x(t),t)}{dt} = -\nabla_x v(x(t),t). \tag{42}$$

To calculate the likelihood of a phase space sample $x$ we map it to the latent space according to Eq.(35) and calculate the jacobian determinant of this transformation according to Eq.(42)

$$r(x) = x + \int_0^1 v_\theta(x,t)dt, \quad \text{with} \quad \left|\frac{\partial r}{\partial x}\right| = \exp\left(\int_0^1 dt \nabla_x v_\theta(x(t),t)\right) \tag{43}$$

$$\Rightarrow \quad p(x) = p_{\text{latent}}(r(x))\exp\left(\int_0^1 dt \nabla_x v_\theta(x(t),t)\right). \tag{44}$$

Solving the ODEs numerically with the required precision takes $\mathcal{O}(100)$ evaluations of the function. For the transformation ODE this is relatively fast as the function is just the velocity, i.e. the neural network. For the likelihood ODE however evaluating the function means calculating the gradients of all components of the velocity with respect to the inputs, making likelihood calculation significantly slower.

The hyperparameters of our CFM network are given in Tab. 2. It is straightforward to replace the Transfer-cINN with a Transfer-CFM in our MEM architecture, so we can benchmark the performance gain through the increased expressivity, at the possible expense of speed.

The likelihoods extracted with the help of the Transfer-CFM are illustrated in Fig. 5 and can be compared to the same MEM setup, but with a Transfer-cINN in Fig. 3. For 400 events the difference between the Transfer-cINN and the Transfer-CFM is not visible, suggesting that both of them work extremely well given the statistical limitations and the phase space integration. There is no systematic bias, and the width of the extracted likelihoods are close to the optimal hard-process curves.

For the high-precision case with 10k events the Transfer-CFM leads to a significant improvement over the cINN architecture. Now, the picture is the same as for 400 events, where the extracted likelihoods do not show any significant bias, and the extracted likelihoods are extremely close to the optimal information.

# 6 Combinatorics transformer

In our last step, we introduce a transformer [43,97,98] to combine the stability and precision of the Transfer-cINN and Transfer-CFM with an appropriate treatment of jet combinatorics [123]. The structure follows the idea that the transfer probability turns a sequence of parton-level momenta into a sequence of reco-level momenta. The Transfer-Transformer, in short Transfermer, should be ideal to encode the correlations between the different particles, without relying on locality or any other physics-inspired requirement.

**Transfermer**

The challenge of using a transformer in our MEM setup is that it is not invertible and does not guarantee a tractable Jacobian. We can solve this problem by making the architecture autoregressive at the level of reco-level momenta and splitting it into two parts, as illustrated in the left panel of Fig. 6: (i) the transformer encodes the correlations between the parton-level and reco-level objects. Their cross-correlation describes the input-output combinatorics;

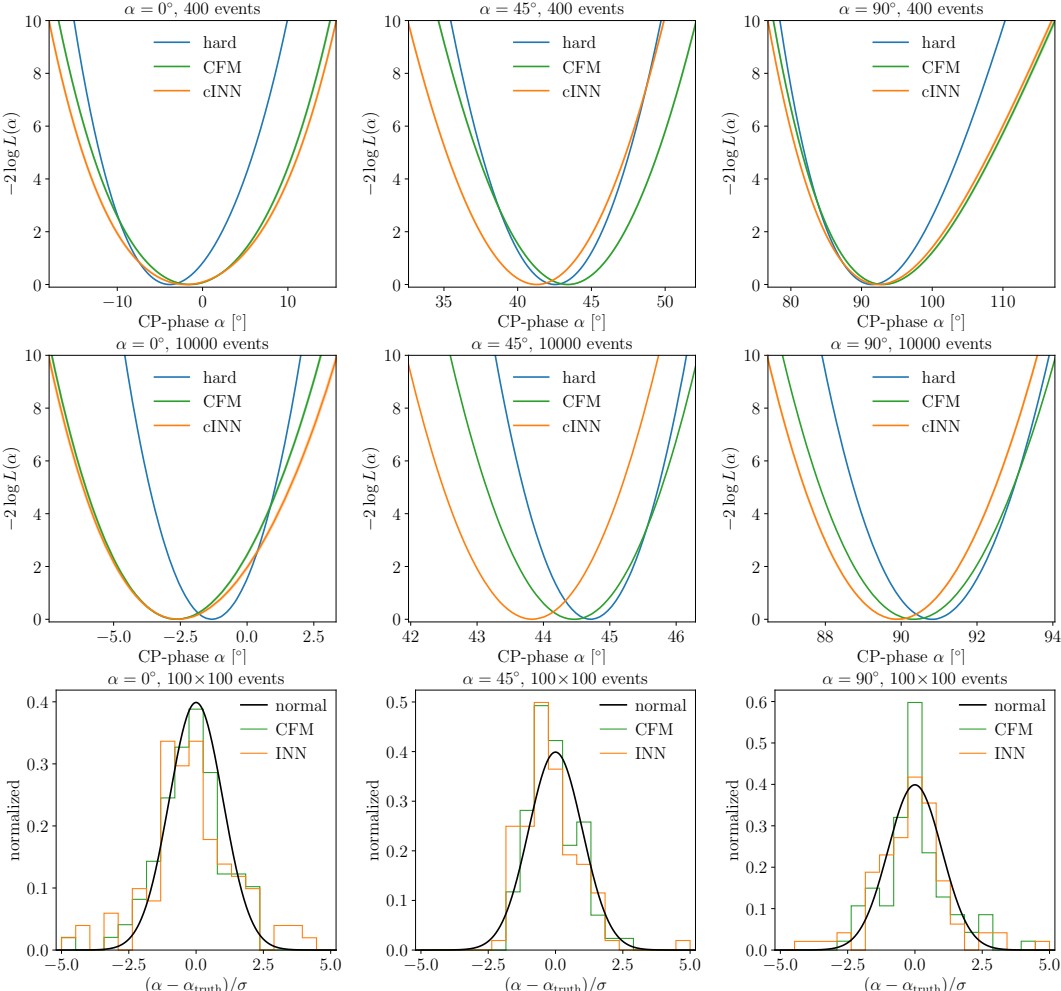

Figure 5: **Transfer-CFM:** likelihoods for different CP-angles. We compare the cINN baseline with a CFM diffusion network, both including the learned acceptance. From top to bottom: likelihoods for 400 events, 10000 events, and pulls.

(ii) a small and universal cINN encodes the correlations between the momentum components of a single particle, conditioned on the output of the transformer $c^{(i)}$.

To guarantee a tractable Jacobian of the full normalizing flow, we apply an autoregressive factorization of the transfer probability defined in Eq.(B.1),

$$p(x_{\text{reco}}|x_{\text{hard}}) = \prod_{i=1}^{n} p(x_{\text{reco}}^{(i)}|c(e_{\text{reco}}^{(0)}, \dots, e_{\text{reco}}^{(i-1)}, e_{\text{hard}})). \tag{45}$$

The function $c$ denotes the transformer encoding. We define a special starting token $e_{\text{reco}}^{(0)}$, shift the inputs by one and mask the self-attention matrix using a triangular mask to ensure that every momentum is only conditioned on the previous momenta. $e_{\text{reco}}^{(i)}$ and $e_{\text{hard}}^{(i)}$ denote the particle-wise embeddings of the momenta and their position. We define this embedding as the concatenation of the momenta and their one-hot-encoded position in the event, padded with zeros. Using a single linear layer instead of the zero-padding does not lead to any performance improvements. We then sample from the transfer probability iteratively, which requires $n$ Transformer evaluations,

$$p(x_{\text{reco}}^{(i)}|x_{\text{hard}}) \equiv p(x_{\text{reco}}^{(i)}|c(e_{\text{reco}}^{(0)}, \dots, e_{\text{reco}}^{(i-1)}, e_{\text{hard}})). \tag{46}$$

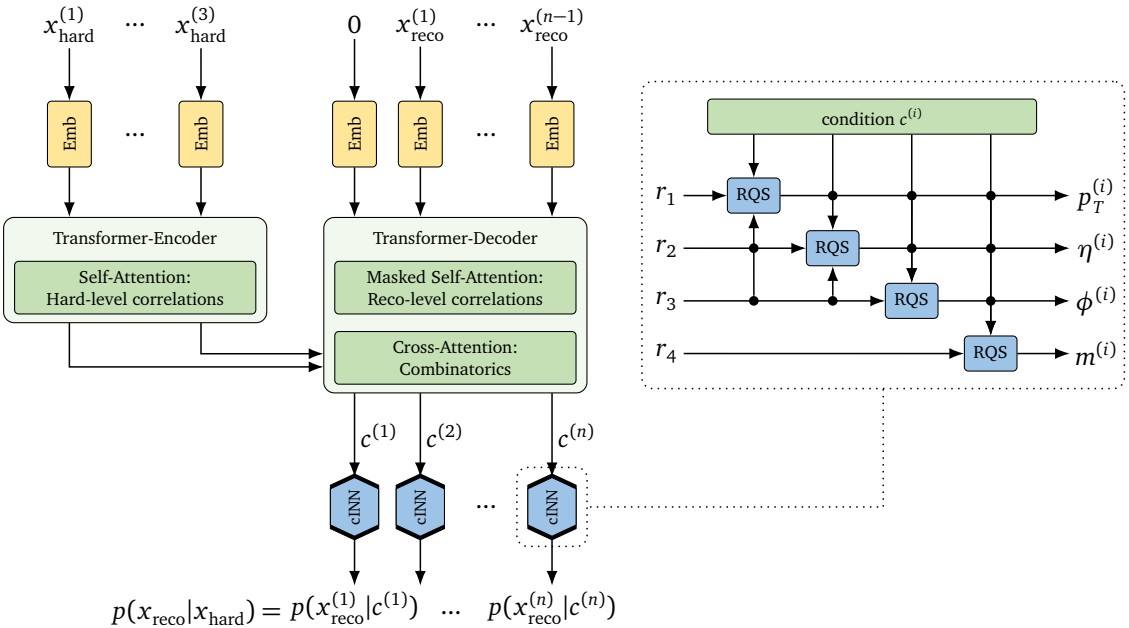

Figure 6: Left: transformer combined with cINN, encoding the transfer probability. Right: cINN used to learn individual momenta, where $r$ is the usual latent space to parametrize a generative model.

Since all $c^{(i)}$ can be computed in a single step from the reco-level momenta, density estimation and training this model is very fast. This is also the way the Transfermer is used during the MEM integration.

The transfer probability in Eq.(45) still has to be converted into a probability distribution for the 4-momentum components of the external particles. To encode massless and massive particles in the same cINN we factorize it into

$$p(x_{\text{reco}}^{(i)}|c^{(i)}) = p(p_T^{(i)}, \eta^{(i)}, \phi^{(i)}|c^{(i)}) \times p(m^{(i)}|p_T^{(i)}, \eta^{(i)}, \phi^{(i)}, c^{(i)}), \qquad (47)$$

such that the generation of the mass direction can be omitted without affecting the other three components. The corresponding cINN architecture is given in the right panel of Fig. 6. Rational quadratic spline coupling layers model the one-dimensional distributions. By transforming each momentum component once and conditioning it on the other components and the transformer output, using a feed-forward network, we build a minimal cINN that is able to model the correlations between the momentum components.

In practice, we use normalized versions of $\log p_T$ and $\log m$ as inputs for the network and map them to Gaussian latent spaces. Similarly, we map $\phi$ and $\eta$ to uniform latent spaces, taking into account the detector-level $\eta$ cuts. For $\phi$ we use periodic RQS splines [88]. The cINN for single momenta and the transformer are trained jointly by minimizing the negative log-likelihood loss $\mathcal{L} = -\log p_\theta(x_{\text{reco}}|x_{\text{hard}})$.

We implement the Transfermer with the standard PYTORCH [124] transformer module and the cINN architecture described above. The hyperparameters are given in Tab. 2. In Fig. 7, we show the likelihoods for the Transfermer architecture. This plot shows much larger error bands because they also include the systematic uncertainty from the Transfermer training, estimated with a Bayesian network. For the other architectures, we omit these due to runtime constraints. The likelihoods can be compared to the cINN results in Fig. 3, and we see that their bias and accuracy have improved. Even for 10k events, the likelihoods are largely unbiased, albeit not significantly better than for the Transfer-CFM from Fig. 5. The Transfermer

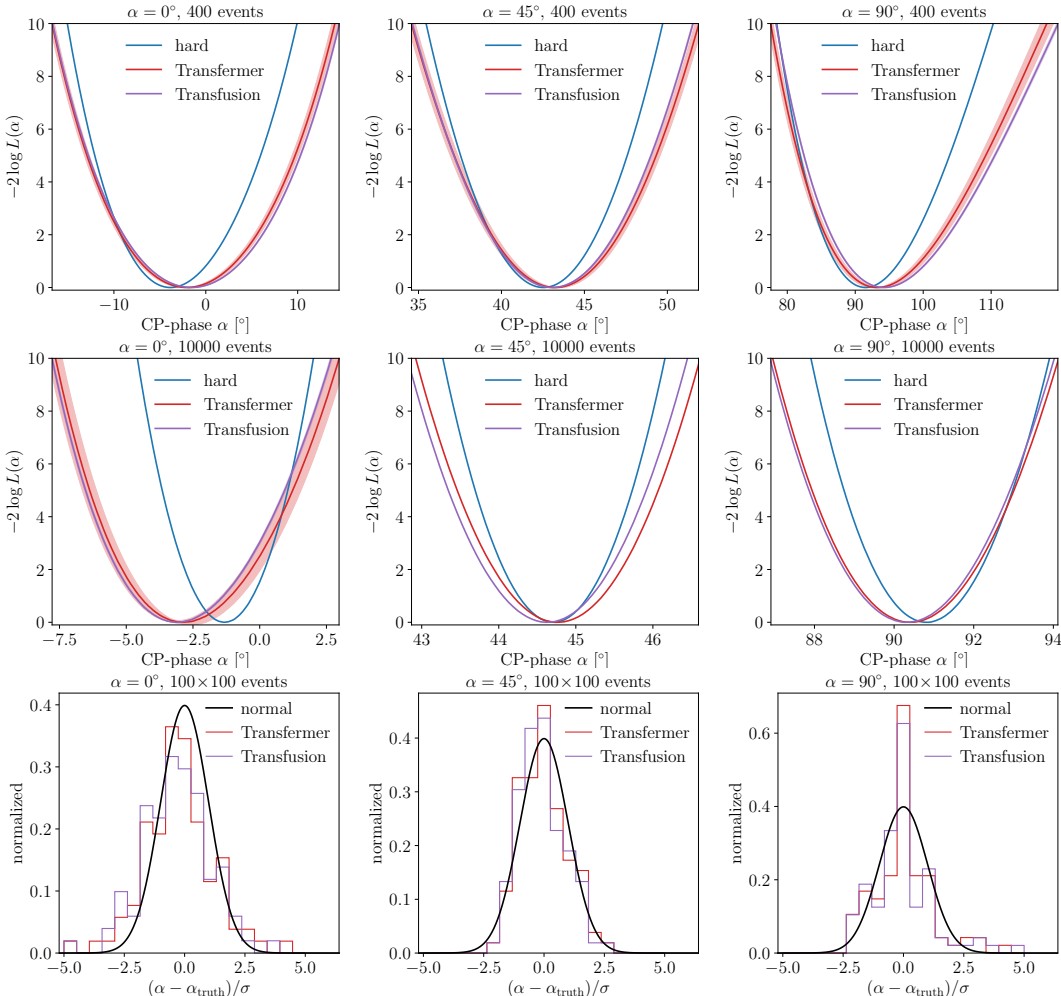

Figure 7: **Transfermer and Transfusion:** likelihoods for different CP-angles using a transformer for the transfer probability, combined with a cINN or a CFM network, respectively. From top to bottom: likelihoods for 400 events, 10000 events, and pulls. Only the Transfermer curve includes the training uncertainties estimated with the Bayesian network.

architecture can be easily generalized to support variable numbers of reco-level jets. We show this extension in Appendix B but do not find any additional improvements for our reference process. Furthermore, we show how sensitive this architecture is to the choice of simulation tool in Appendix C.

**Transfusion**

As a last transfer architecture we consider the CFM equivalent of the Transfermer, an autoregressive Transfusion. We keep the autoregressive structure and the masked self-attention from Fig. 6 and simply replace the small cINN with a small CFM network to generate the individual particle momenta. The CFM learning task is a simple regression of the velocity field. As long as we can track gradients through the network, we obtain a tractable Jacobian according to Eq.(42). The velocity of the $i^{\text{th}}$ particle is then denoted in analogy to Eq.(45)

$$v^{(i)}(x_{\text{reco}}^{(i)}(t), t | c(e_{\text{reco}}^{(0)}, \dots, e_{\text{reco}}^{(i-1)}, e_{\text{hard}})), \tag{48}$$

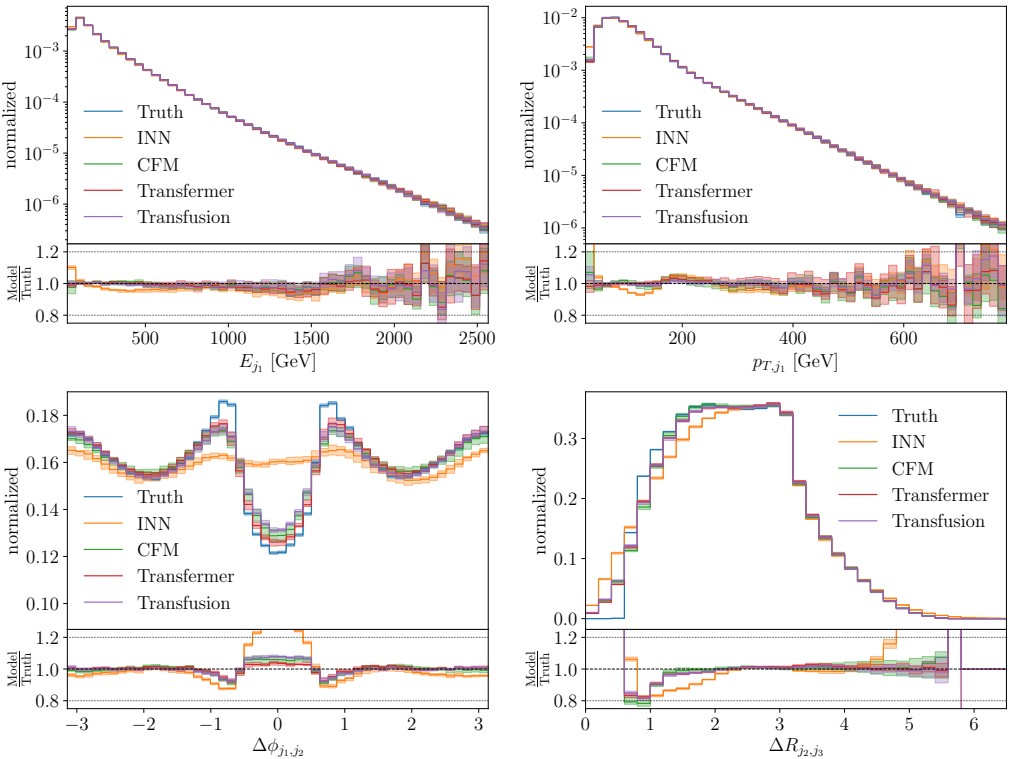

Figure 8: Reco-level distributions for different kinematic observables, obtained from the different generative transfer networks, conditioned on the hard-scattering momenta. Truth corresponds to the high-statistics training data.

From the velocity field the likelihoods are again obtained by solving the ODEs Eq.(44), in the Transfusion setup now autoregressively for each particle. For on-shell and off-shell particles, we use two different small CFMs, one 3-dimensional and one 4-dimensional. This setup outperforms just using the same 4-dimensional network and discarding the generated masses for on-shell particles. The hyperparameters of the Transfusion network are given in Tab. 3.

We show the MEM likelihoods obtained with the Transfusion in Fig. 7 and find that they are indistinguishable from the Transfermer results. This indicates that the difference between the cINN and CFM likelihoods can be attributed to cINN issues with the jet combinatorics. Outsourcing this task to the transformer significantly improves the performance. For the CFM the corresponding improvement is minimal.

# 7 Outlook

The matrix element method is an example of an LHC inference method, which is hugely attractive but only enabled by modern machine learning [21]. Specifically, it requires a fast and precise forward-transfer probability, an extremely efficient phase space mapping for the integration over the hard phase space, and a flexible encoding of the detector efficiency. We have shown, for a CP measurement in the associated production of a Higgs and a single top, how each of these tasks can be assigned to a neural network. This combination of three networks with modern architectures provides the required precision and speed.

To illustrate the performance of the different network architectures and MEM frameworks, we show a set of kinematic observables from the generative transfer networks in Fig. 8. Before integrating the likelihood, we can show the distributions at the reco-level and compare them to

the truth, or training data. We immediately see that the standard cINN is stable and extremely fast, but limited in its expressivity. The CFM diffusion network improves the performance significantly. The transformer architectures, i.e. the cINN-driven Transfermer and the CFM-driven Transfusion deliver a precision at least on par with the CFM diffusion network.

For the likelihood, we have compared the extracted likelihoods from the different architectures with the hard-process target. As a benchmark, we first improved a range of numerical aspects of our concept paper [21], with a focus on the integration with the Sampling-cINN. The improved precision of the integration raises two questions: a systematic bias in the minimum of the extracted likelihoods especially going from 400 to 10k events; and the optimality of the extracted likelihoods seen in the widths in the CP-angle $\alpha$. These benchmark results are shown in Fig. 3.

We then upgraded our two-network setup to a three-network setup, with a learned acceptance as a function of phase space. In Fig. 3, we saw that this removes the leading source of systematic bias, including the challenging SM-case $\alpha = 0°$.

Next, we targeted the performance of the transfer network by replacing it with a more expressive, albeit slower CFM diffusion network. This did not improve the low-statistics results, but for the high-statistics case of 10k events the Transfer-CFM showed a clear advantage over the cINN, as can be seen in Fig. 5.

Finally, we solved the problem with the jet multiplicity of the cINN approach by applying a generative autoregressive Transfer-Transformer, i.e. combining a transformer with a cINN network (Transfermer) and a CFM-model (Transfusion). In Fig. 7, we saw that both transformer-based models outperformed the cINN, but showed similar performance as the Transfer-CFM. Notably, both transformer-based models can naturally be extended to describe a variable number of particles at both reco- and parton-level. This feature will eventually be needed for a proper description of the MEM at NLO.

In our LO example, all three models, CFM, Transfermer and Transfusion, parametrize the transfer probability flexibly and reliably. However, the Transfermer integration is approximately a factor 30 faster than the two diffusion-based models. This gap might eventually be closed using techniques like diffusion distillation [125–127]. Further improvements on the architecture, like the parallel Transfusion introduced in Appendix B, might also improve the performance for more complex processes. Altogether, we conclude that a range of modern generative networks are available for the MEM, awaiting final judgment from an actual analysis.

## Acknowledgments

**Funding information** We would like to thank the Baden-Württemberg-Stiftung for funding through the program *Internationale Spitzenforschung,* project *Uncertainties — Teaching AI its Limits* (BWST_IF2020-010). AB and TP are supported by the Deutsche Forschungsgemeinschaft (DFG, German Research Foundation) under grant 396021762 – TRR 257 *Particle Physics Phenomenology after the Higgs Discovery.* AB, and NH are funded by the BMBF Junior Group Generative Precision Networks for Particle Physics (DLR 01IS22079). TH is funded by the Carl-Zeiss-Stiftung through the project *Model-Based AI: Physical Models and Deep Learning for Imaging and Cancer Treatment.* RW acknowledges support by FRS-FNRS (Belgian National Scientific Research Fund) IISN projects 4.4503.16. The authors acknowledge support by the state of Baden-Württemberg through bwHPC and the German Research Foundation (DFG) through grant no INST 39/963-1 FUGG (bwForCluster NEMO). Computational resources have been provided by the supercomputing facilities of the Université catholique de Louvain (CIS-M/UCL) and the Consortium des Équipements de Calcul Intensif en Fédération Wallonie Brux-

elles (CÉCI) funded by the Fond de la Recherche Scientifique de Belgique (F.R.S.-FNRS) under convention 2.5020.11 and by the Walloon Region. This work was supported by the DFG under Germany's Excellence Strategy EXC 2181/1 - 390900948 *The Heidelberg STRUCTURES Excellence Cluster.*

## A  Network hyperparameters

Table 1: Hyperparameters of the classifiers learning the acceptance $\epsilon(x_{\text{hard}})$ (left) and the jet multiplicity used in Appendix B (right).

| Parameter | Acceptance | Multiplicities |
|---|---|---|
| Optimizer | Adam | |
| Learning rate | 0.0001 | |
| LR schedule | One-cycle | |
| Maximum learning rate | 0.0003 | |
| Batch size | 1024 | |
| Epochs | 10 | |
| Number of layers | 6 | |
| Hidden nodes | 256 | |
| Activation function | ReLU | |
| Preprocessing | $p_T, \eta, \phi, m$ | |
| Loss | Binary cross-entropy | Categorical cross-entropy |
| Training samples | 5M | 3.4M |
| Validation samples | 500k | 340k |
| Testing samples | 4.5M | 3.1M |
| Trainable parameters | 266k | 266k |

Table 2: Hyperparameters of the CFM (left) and the Transfermer (right).

| Parameter | Value | Parameter | Value |
|---|---|---|---|
| Optimizer | Adam | Optimizer | RAdam |
| Learning rate | 0.001 | Learning rate | 0.0001 |
| LR schedule | Cosine-annealing | LR schedule | One-cycle |
| Batch size | 16384 | Maximum learning rate | 0.0003 |
| Epochs | 1000 | Batch size | 1024 |
| Number of layers | 8 | Epochs | 200 |
| Feed-forward dimension | 512 | Number of heads | 8 |
| Activation function | SiLU | Number of encoder layers | 6 |
| Training samples | 3.4M | Number of decoder layers | 8 |
| Validation samples | 340k | Embedding dimension | 64 |
| Testing samples | 3.1M | Transformer feed-forward dimension | 256 |
| Trainable parameters | 3.2M | Number of subnet layers | 5 |
| ODE solver method | Runge-Kutta 4 | Subnet hidden nodes | 256 |
| Solver step-size | 0.05 | Subnet activation function | ReLU |
| | | RQS spline bins | 16 |
| | | Training samples | 3.4M |
| | | Validation samples | 340k |
| | | Testing samples | 3.1M |
| | | Trainable parameters | 2.6M |

Table 3: Hyperparameters of the autoregressive Transfusion (left) and the parallel Transfusion (right).

| Parameter | Value | Parameter | Value |
|---|---|---|---|
| Optimizer | Adam | Optimizer | Adam |
| Learning rate | 0.001 | Learning rate | 0.001 |
| LR schedule | Cosine-annealing | LR schedule | Cosine-annealing |
| Batch size | 8192 | Batch size | 8192 |
| Epochs | 600 | Epochs | 600 |
| Number of heads | 8 | Number of heads | 4 |
| Number of encoder layers | 6 | Number of encoder layers | 6 |
| Number of decoder layers | 8 | Number of decoder layers | 6 |
| Embedding dimension | 64 | Embedding dimension | 128 |
| Transf. feed-forward dim | 256 | Transf. feed-forward dim | 512 |
| Number of layers CFM | 6 | Training samples | 3.4M |
| Hidden nodes CFM | 400 | Validation samples | 340k |
| Activation function CFM | ReLU | Testing samples | 3.1M |
| Training samples | 3.4M | Trainable parameters | 2.7M |
| Validation samples | 340k | ODE solver method | Runge-Kutta, order 4 |
| Testing samples | 3.1M | Solver step-size | 0.05 |
| Trainable parameters | 3.5M | | |
| ODE solver method | Runge-Kutta, order 4 | | |
| Solver step-size | 0.05 | | |

# B   Variable jet number and permutation invariance

**Transfermer with variable jet number**

The Transfermer is easy to generalize to events with a variable number of jets at the reconstruction level. To this end, we split the inclusive transfer probability and evaluate it autoregressively,

$$p(x_{\text{reco}}, n | x_{\text{hard}}) = p(n | x_{\text{hard}}) \, p(x_{\text{reco}} | x_{\text{hard}}, n)$$

$$= p(n | x_{\text{hard}}) \, p(x_{\text{reco}}^{(1:n_{\min})} | x_{\text{hard}}, n) \prod_{i=n_{\min}+1}^{n} p(x_{\text{reco}}^{(i)} | x_{\text{reco}}^{(1:i-1)}, x_{\text{hard}}, n), \qquad \text{(B.1)}$$

where $n$ is the number of final-state particles, $x_{\text{reco}}^{(1:k)}$ denotes the first $k$ reco-level momenta, $x_{\text{reco}}^{(k)}$ denotes the $k$-th reco-level momentum and $n_{\min}$ is the minimal number of momenta for an accepted event. The probability $p(n | x_{\text{hard}})$ can be extracted using a simple classifier network with a categorical cross-entropy loss and the number of additional jets as labels. The autoregressive factorization of $p(x_{\text{reco}} | x_{\text{hard}}, n)$ matches the way in which the Transfermer learns these probabilities. We pass the information about the number of additional jets to the Transfermer by appending it to the embedding of $x_{\text{hard}}$ in one-hot encoded form. We can sample from the transfer probability by first sampling the multiplicity using the probabilities given by the classifier and then sampling the momenta as described in Eq.(46). Note that it is even possible to generalize the Transfermer to a variable number of hard-scattering momenta, because the transformer encoder accepts a variable number of inputs without any further changes to the architecture, making it a good candidate for a machine-learned MEM at NLO.

We train the jet multiplicity classifier with the hyperparameters given in Tab. 1. We observe that they are mostly flat for the top and Higgs, but there is a stronger variation as a function of the forward jet momentum, especially $\eta_j$. Like for the acceptance function, this is explained by ISR jets being tagged instead of the forward jet, leading to a lower probability of extra jets for $|\eta| > 2.4$.

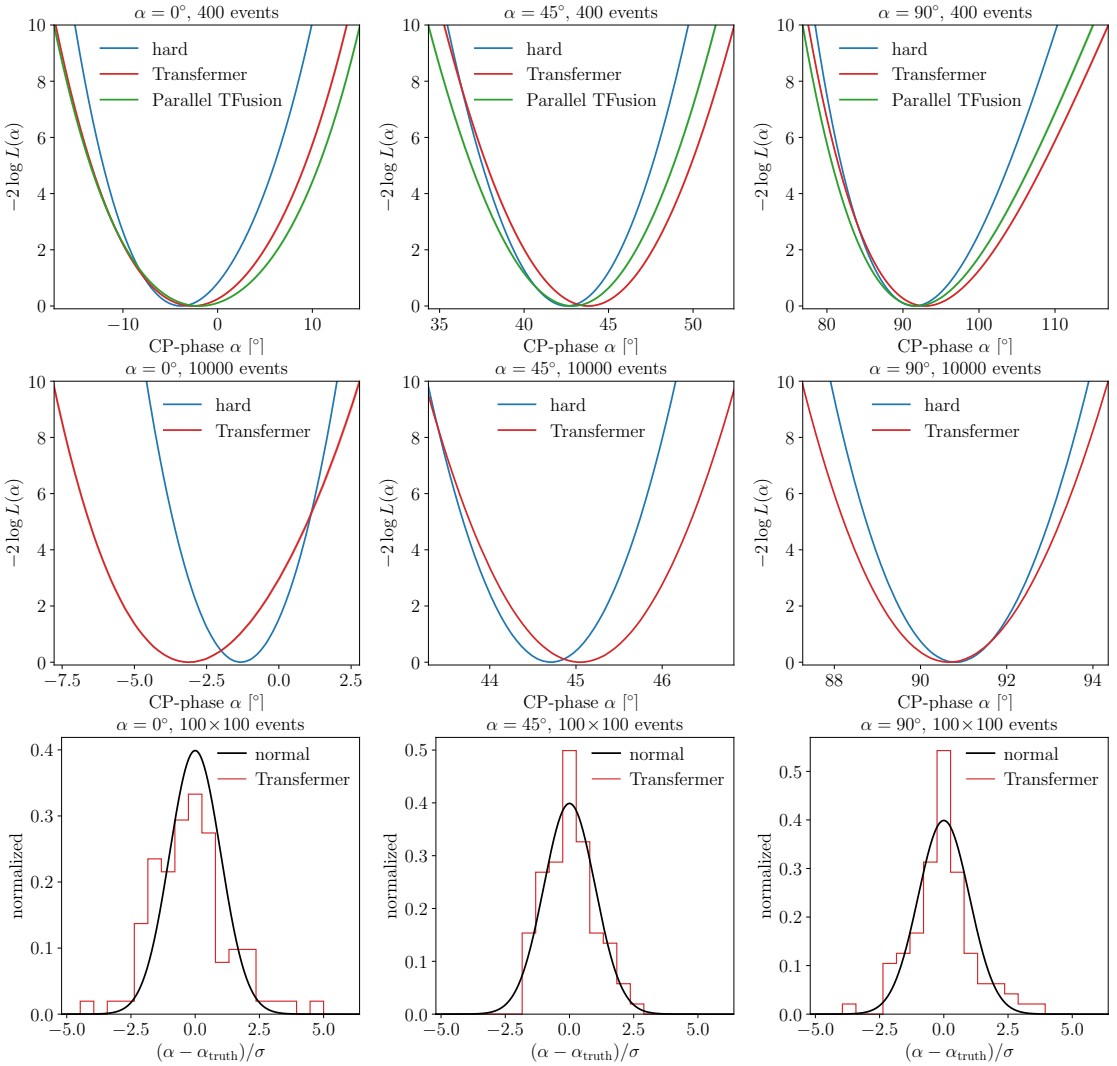

Figure 9: **Transfermer with variable jet numbers and parallel Transfusion:** likelihoods for different CP-angles using the Transfermer with variable jet multiplicity and the parallel Transfusion as the transfer probability. From top to bottom: likelihood for 400 events, 10000 events, and pulls.

We then run the MEM integration to obtain the results shown in Fig. 9. They are mostly similar to the results with fixed multiplicity shown in Fig. 7. It shows that for our specific process, we do not gain constraining power by including the information from additional jets. However, that might be different for other processes and especially at NLO. So the ability to deal with a variable number of jets is still a valuable addition to our MEM toolbox.

**Permutation-invariant transfusion**

The Transfusion can be generalized to events with a variable jet number in complete analogy to the Transfermer. However, as diffusion models do not require invertibility, they allow for an additional approach in combining the transformer with the CFM network where we drop the autoregressive setup and instead generate all particle 4-momenta in parallel.

Before, in the autoregressive setup the transformer calculates a condition based on the hard-level momenta and the already generated reco-level momenta, which is then fed to the CFM that predicts the time-dependent velocity field. Crucially, the transformer itself has no

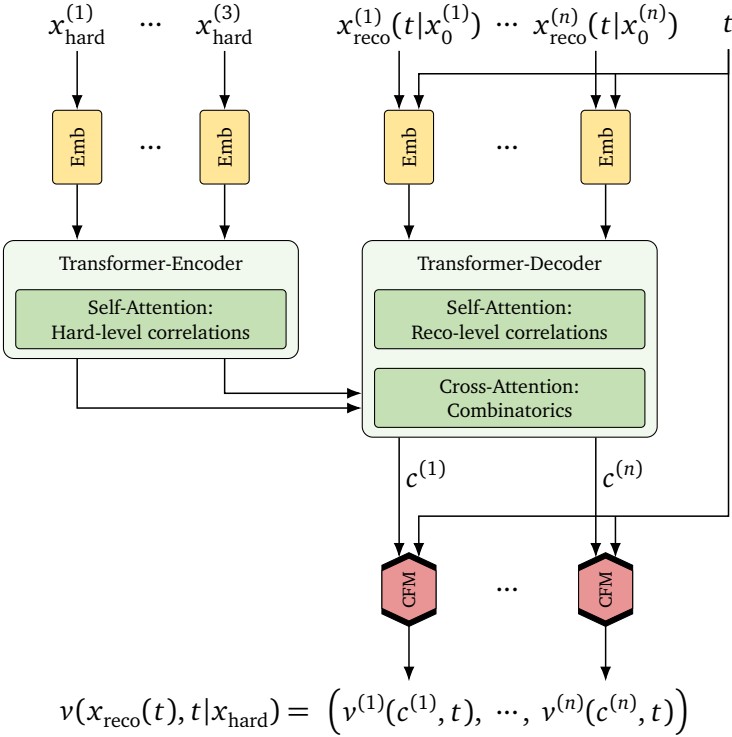

Figure 10: Parallel Transfusion architecture. Compared to the autoregressive setup we no longer use masked self-attention in the transformer decoder, but instead make it time-dependent.

time dependence. In the alternative parallel setup, the transformer decoder no longer sees the first $i-1$ reco-level particles $x_{\text{reco}}^{(1,\dots,i-1)}$ to describe $c^{(i)}$. Instead, its inputs are the conditional and time-dependent diffusion states $x_{\text{reco}}^{(1,\dots,n)}(t|x_0)$, as defined in Eq.(38), of all $n$ reco-level particles, and the time t. The encoder, which acts only on the hard-level momente, is unchanged. Now, the transformer calculates time-dependent embeddings, one for each particle. These time-dependent embeddings are then again fed to a small CFM network predicting the velocity field. In this setup the velocity field of the $i^{\text{th}}$ particle is calculated as

$$v^{(i)}(c(e_{\text{reco}}(t), e_{\text{hard}}, t), t),\tag{B.2}$$

where the transformer c is now a time-dependent function of the embeddings $e$ of all momenta. The overall setup is illustrated in Fig. 10. In practice, a single linear layer is sufficient to map the transformer outputs to the velocity field components. Note, that during sampling the initial input to the transformer is the unconditional latent space vector $r$ which is then mapped onto $x_{\text{reco}}$ with the learned velocity field and the ODE solver. The parallel Transfusion setup naturally generalizes to varying particle multiplicities at both hard- and reco-level without requiring an arbitrary autoregressive order, as it is permutation-invariant at both levels.

Reco-level distributions for different kinematic observables are shown in Fig. 11. The marginal distributions show no difference between the parallel Transfusion and the other networks, but for the angular correlations we see the parallel Transfusion having a clear edge. Giving the transformer itself a time-dependence forces us to evaluate it repeatedly inside the ODE solver, making sampling and likelihood calculation in this setup even slower than for the pure CFM or the autoregressive Transfusion. We show integration results for 400 events using the parallel Transfusion in Fig. 9, finding that they are comparable to the results from the autoregressive Transfusion. Due to the slow likelihood calculation this setup did not scale up

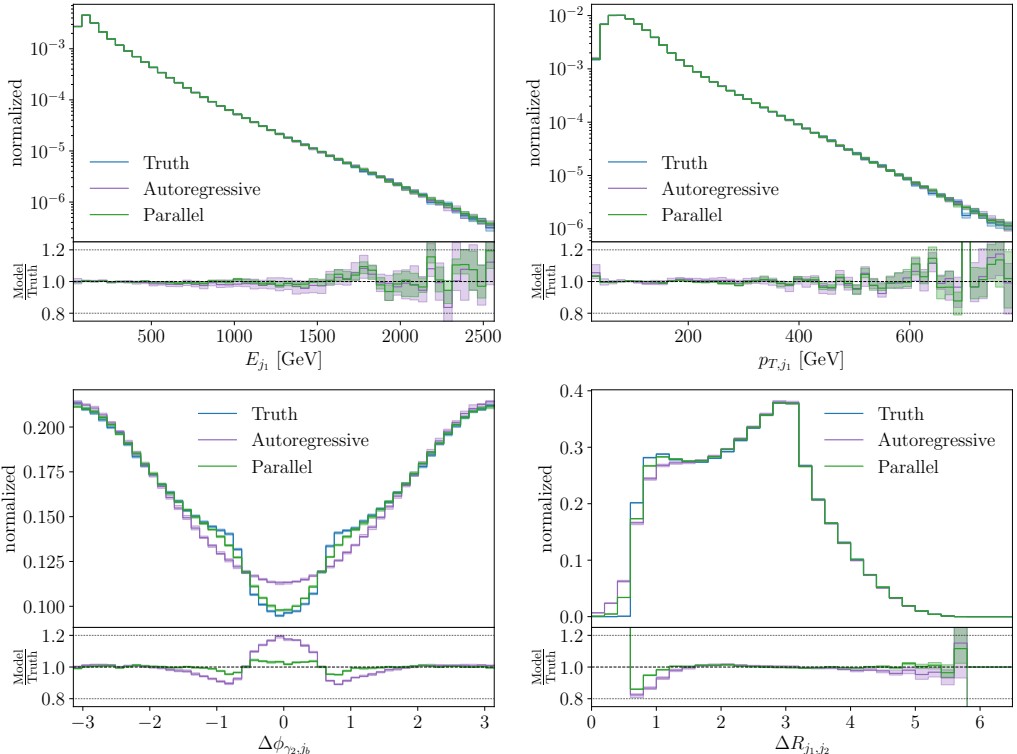

Figure 11: Reco-level distributions for different kinematic observables, obtained from the autoregressive and parallel Transfusion networks, conditioned on the hard-scattering momenta. Truth corresponds to the high-statistics training data.

to 10000 events. The strong performance on the observable distribution level indicates that this architecture might proof useful in combination with speed-up techniques like diffusion distillation or for applications that do not require likelihood calculation.

## C  Evaluating on Herwig

The critical backbone of our inference method is the learned transfer probability $p_\theta(x_{\text{reco}}|x_{\text{hard}})$. We have demonstrated that generative networks can learn this conditional density from simulated data to very high precision. However, even a perfect network will only encode the forward transfer of the simulation, which is close but not necessarily identical to nature. In this section, we investigate how this impacts the results of our method by using different simulation setups:

1. a baseline simulation with PYTHIA for network training as described in Sec. 1;

2. an alternative simulation based on HERWIG [128] for inference, emulating the truth reco-level data of the experiment. The detector effects are still modeled with DELPHES.

The results obtained with our method in this setup are shown in Fig. 12. For 400 events we find that the extracted reco-level likelihoods mostly agree with the hard-level likelihoods. Note that the hard-level likelihoods are not fixed but also affected by the underlying simulation assumption, most visible for $\alpha = 90°$. This is because the fiducial hard-level likelihood is only defined on hard-level events $x_{\text{hard}}$ leading to accepted $x_{\text{reco}}$ events, which critically depends on the efficiency $\epsilon(x_{\text{hard}})$ of the underlying normalized transfer function $r$, as defined

in Eqs.(5) and (12). This effectively encodes a dependence on the assumed forward simulation

$$p_{\text{fid}}(x_{\text{hard}}|\alpha) \equiv p_{\text{PYTHIA}}(x_{\text{hard}}|\alpha). \tag{C.1}$$

Hence, evaluating the fiducial hard-level likelihoods on the HERWIG simulation can generally lead to a bias in the likelihood distribution. In the high-statistics scenario with 10k events, we observe good agreement for $\alpha = 0, 45°$, comparable to the results when evaluating on PYTHIA, which means we can assume

$$p_{\text{PYTHIA}}(x_{\text{hard}}|\alpha) \approx p_{\text{HERWIG}}(x_{\text{hard}}|\alpha). \tag{C.2}$$

In these cases, we find that the reco-level likelihood obtained using our method still agrees well with the hard-scattering likelihood, and the results are still well-calibrated. However, for $\alpha = 90°$ the hard-level likelihoods are significantly off from the true value, indicating that Eq.(C.2) is no longer valid. Further, training the transfer function on events that do not follow the true distribution of the measured data may introduce $\alpha$-dependent effects. Consequently, we also find a large deviation between the hard- and reco-level likelihoods.

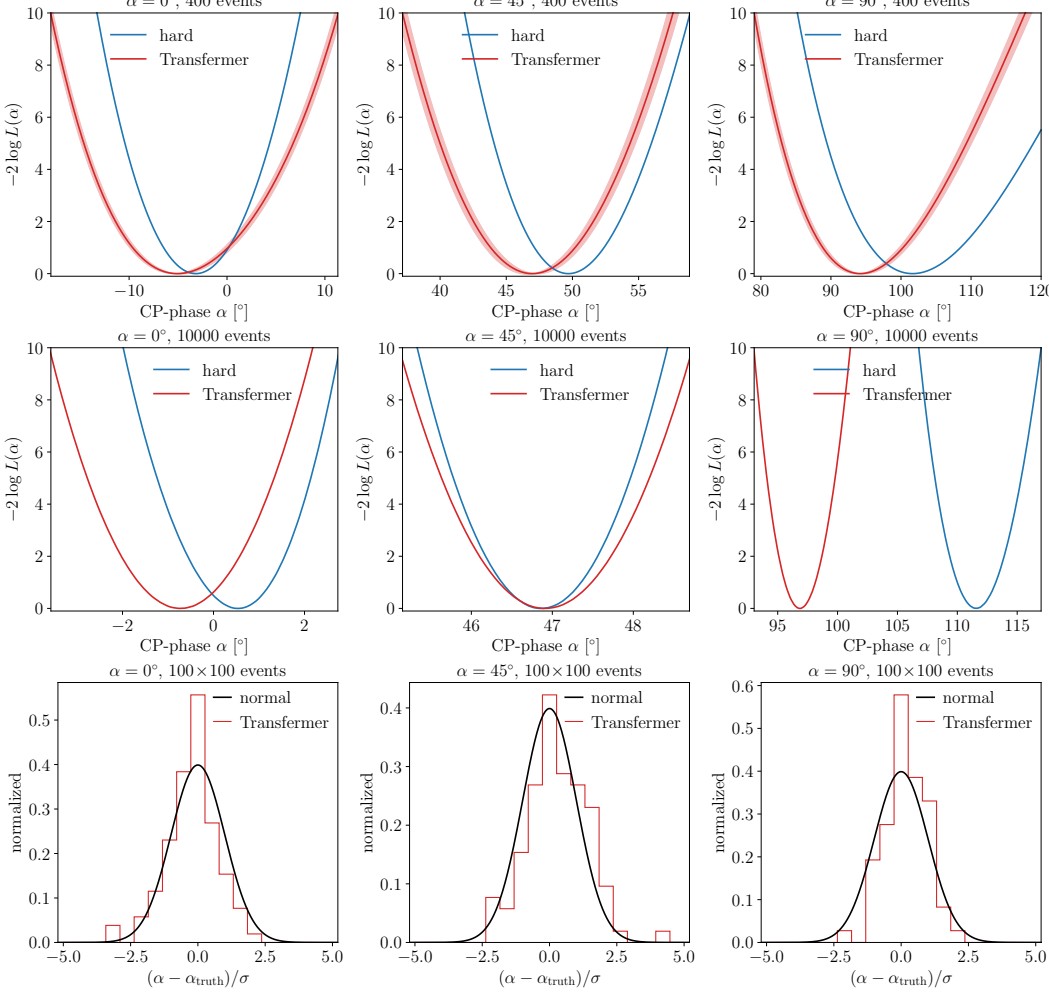

Figure 12: **Transfermer applied on HERWIG simulation:** likelihoods for different CP-angles using the Transfermer trained on PYTHIA simulations but evaluated on HERWIG simulations. From top to bottom: likelihood for 400 events, 10000 events, and pulls.

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
