# Peer review of "Precision-Machine Learning for the Matrix Element Method"

_SciPost Physics, doi:SciPost Phys. 17, 129 (2024)_

## Round 2 · Referee Report · Anonymous (Referee 1) · 2024-1-30

Strengths

  1. Explores an innovative approach for extracting fundamental parameters of the underlying Lagrangian from an event sample
  2. Contains a concise description of the machine learning methods applied

Weaknesses

  1. The description of the method used for generating the event sample is lacking
  2. It is unclear whether the results of the developed method relies on training of the network and the check of the extraction relies on both data sets being generated by the same method.

Report

This submission is a well presented study developing machine learning techniques to extract parameters of the underlying Lagrangian from a generated event sample using the matrix element method. The presentation is concise and well written, with only a few typos (e.g. "as well"->"as well as" in the second line of the introduction). Some of the acronyms used are not defined (e.g."cINNs" in the last paragraph of the introduction).

More importantly though, there is no description of how the event sample was generated - this is important, since it is unclear whether the try complexity of the process has been described correctly (and therefore whether the underlying parameter \alpha has been too easy to extract for real-world applications to be relevant). Was the process modelled with a parton shower? At what perturbative precision? (Born or NLO+shower, merged MEPS?). A description of the choices made for the generation would need to be added.

Furthermore, it would seem prudent to check to what extent the conclusions on the usefulness of the method depends on the training and the checks being performed on the same sample (or samples generated with the same choices, so differing only by statistics).

Requested changes

  1. describe the method used for generating the event sample(s)
  2. check the quality of the extracting for a sample generated with the same choice for couplings, but using a different generator (Herwig/Sherpa/Pythia) or perturbative matching (CKKWL with high-multiplicity matching/MC@NLO etc).

  • validity: good
  • significance: high
  • originality: good
  • clarity: high
  • formatting: excellent
  • grammar: good

Author:  Ramon Winterhalder  on 2024-10-04  [id 4832]

(in reply to Report 2 on 2024-01-30)

Dear referee,

We would like to thank you for your detailed and careful comments on our manuscript, which helped us to improve the presentation of our research results.

Below we give some detailed answers to the various points raised by:

  1. describe the method used for generating the event sample(s)

As mentioned in the last paragraph before Section 2, the details of the reference process, including the method to generate the events, have been described in the previous paper [arXiv:2210.00019], which we cite here. In short, we rely on a toolchain based on MadGraph5 (parton-level), Pythia (parton shower), Delphes (fast detector simulation), and FastJet (jet reconstruction).

  1. check the quality of the extracting for a sample generated with the same choice for couplings, but using a different generator (Herwig/Sherpa/Pythia) or perturbative matching (CKKWL with high-multiplicity matching/MC@NLO etc).

We have added an alternative evaluation of the likelihood based on performing the inference on data simulating the parton showers with Herwig. The results and discussion are shown in Appendix C in the new version.

---

## Round 2 · Referee Report · Simon Badger (Referee 2) · 2024-2-1

Report

The article contains a detailed description and analysis of a proof-of-concept machine learning (ML) based implementation of the matrix element method which is widely used in experimental analyses. The article follows on from a previous study (arXiv:2210.00019, with some overlap with the current author list) which employed invertible neural networks and introduces a new, more refined, setup with a variety of state-of-the-art generative and transformer ML models.

The authors use the same example as the previous study which aids comparison to that setup.

  1. The word 'precision' in the title was slightly confusing at first. Since there are many contexts in this subject in which precision is a major issue perhaps it is worth clarifying in the abstract the a more specific reference to the quantity in this article that is denoted to be precise.

  2. The acronym 'cINN' (or INN for that matter) is not defined at the first usage in the introduction.

  3. The description of the ML-matrix element method (sec 2) follows that of Ref. [21] . It would probably benefit the reader is the section would start with a reference to this article and a summary any major updates/changes that are introduced in this article.

I believe it would be beneficial for readers to have a clearer description of the notation in particular for $\chi_{\rm hard}$ and the meaning of the '$\sim$' symbol e.g $\chi \sim p(\chi_{\rm hard}|\alpha)$.

  1. It is my understanding that this article addresses a limitation of the previous ML matrix element method set up presented in arXiv:2210.00019. The authors finish the article by suggesting that application to an actual analysis is necessary before final conclusions of the readiness of this technology. It would be useful if the authors could expand on this point and indicate what sort of features such an analysis would put to the test and which future analyses would be best suited to this approach.

Once the following suggestions have been addressed I believe the article would be suitable for publication in SciPost physics.

---

## Editorial Decision

published